# Determinants of secondary prophylaxis for childhood rheumatic heart disease in Ethiopia: A qualitative study of children and caregivers

Eshetie Melese Birru [1,2,3]*, Kevin T. Batty[1,4], Laurens Manning[2,5], Stephanie L. Enkel[2], Brioni R. Moore[1,2,4]

1 Curtin Medical School, Curtin University, Bentley, Perth, Western Australia, 2 Wesfarmers Centre of Vaccines and Infectious Diseases, The Kids Research Institute Australia, Nedlands, Perth, Western Australia, 3 Department of Pharmacology, School of Pharmacy, College of Medicine and Health Sciences, University of Gondar, Gondar, Ethiopia, 4 Curtin Health Innovation Research Institute, Curtin University, Bentley, Perth, Western Australia, 5 Medical School, The University of Western Australia, Crawley, Perth, Western Australia

* eshetie.birru@thekids.org.au, meshetie21@gmail.com

## Abstract

### Background

Rheumatic heart disease (RHD) remains a major public health challenge in Ethiopia, particularly among children. Monthly intramuscular benzathine penicillin G (BPG) is the cornerstone of secondary prophylaxis, yet adherence and delivery are suboptimal. This study explores the lived experiences of children and adolescents with RHD, capturing their direct voices and those of their caregivers to understand determinants of secondary prophylaxis uptake.

### Methods

A qualitative descriptive study was conducted from May to July 2022 in four Ethiopian public hospitals providing RHD care. Focus group discussions were held with children living with RHD (C-RHD) and their caregivers to explore barriers and facilitators of BPG delivery. Audio recordings were transcribed, translated, and thematically analysed using the framework method, with findings mapped onto the Capability, Opportunity, Motivation, Behaviour (COM-B) model.

### Results

Forty-two participants (30 C-RHD, 12 caregivers) identified five interrelated determinants of BPG delivery: structural barriers (geographic inaccessibility, transport costs, limited policies), organizational challenges (drug shortages, inadequate follow-up and counselling), therapeutic issues (painful injections, needle blockages, limited alternatives), provider-related concerns (fear of adverse reactions), and individual factors (misconceptions, psychological distress, adherence behaviours). Despite a

**Data availability statement:** Due to the sensitive nature of the qualitative interview data and the need to protect participant confidentiality, the raw interview transcripts are not publicly available, as they may contain potentially identifiable information. Requests regarding data access may be directed to the Curtin University Human Research Ethics Committee, referencing approval number HRE2022-0221, at hrec@curtin.edu.au.

**Funding:** The author(s) received no specific funding for this work.

**Competing interests:** No authors have competing interests.

preference for injectable BPG, some children received less effective oral antibiotics due to poor access. Participants emphasized the need for reliable BPG supply, dedicated providers, and strengthened patient support programs.

## Conclusions

Strengthening patient/carer education, improving BPG formulations and access, and addressing resource gaps within healthcare systems are critical to enhancing adherence, preventing ARF/RHD, and improving patient outcomes.

## Background

Acute rheumatic fever (ARF) is an inflammatory autoimmune disease resulting from Group A Streptococcal infections of the throat or skin [1]. Recurrent, untreated episodes of ARF increase the risk of developing rheumatic heart disease (RHD) and, if persistent, permanent heart valve damage that may require surgical intervention [2]. Although streptococcal skin and throat infections are generally responsive to penicillin, RHD remains a neglected and significant public health issue due to gaps in prevention strategies and inadequate support for affected people [3,4].

An estimated 40.5 million people worldwide are living with RHD, resulting in over 300,000 deaths in 2019 [5]. ARF is most associated with children, adolescents and young adults, with peak incidence occurring between 5 and 14 years. Whilst the prevalence of RHD was previously more commonly associated with patients in their 2nd to 4th decade of life [6] recent disease trends have seen a 41% increase in the number of children <15 years of age with RHD (1990–2019) [7]. Socioeconomically disadvantaged individuals in low- and middle-income nations within Oceania, South Asia and sub-Saharan Africa, and marginalized communities in developed countries (e.g., Aboriginal, Torres Strait Islander peoples and immigrant populations within Australia, Māori and Pacific peoples in New Zealand, and Indigenous populations in Fiji), are at highest risk [1,8–12].

Secondary antibiotic prophylaxis is the most cost-effective strategy for controlling ARF and RHD, as it prevents recurrent ARF episodes and the progression of RHD through targeted control programs [13–15]. Intramuscular benzathine penicillin G (BPG) is the cornerstone of secondary prophylaxis for ARF/RHD, as endorsed by the World Health Organization [13]. Administered every 3–4 weeks for at least 5 years following the most recent ARF episode, BPG is more effective than oral antibiotics (e.g., erythromycin, clindamycin, or amoxicillin) in preventing streptococcal infections and reducing ARF recurrence [16]. This approach helps prevent further valvular damage, allows for potential disease regression [17,18], and reduces mortality from RHD [19].

However, it is widely acknowledged that in addition to concerns with suboptimal disease diagnosis [20,21], a large proportion of people with ARF/RHD do not adhere to the recommended monthly BPG injections [22,23]. Qualitative studies from Uganda and Sudan identified some factors influencing the successful delivery

and uptake of secondary prophylaxis for RHD, categorized into health system, intervention, individual, and provider-related factors [24–26]. Additionally, a report from Australia highlighted the complexity of RHD care interventions and advocated for a patient-centred chronic care approach [27]. Investigating and addressing the contributing factors through robust improvement strategies is essential for enhancing national and global initiatives focused on the control and eradication of RHD.

Ethiopia is characterized by one of the highest global burdens of RHD, with an estimated prevalence surpassing 3% [28–31]. There is currently no national strategy for RHD prevention, control, or treatment [32]. Although guidelines recommend monthly administration of BPG for secondary prophylaxis [33], delivery to vulnerable populations remains variable and suboptimal and often impacted by distance of travel to healthcare facilities, financial constraints, and affordability [32,34,35]. Prior qualitative studies conducted in Uganda and other African settings [24,26], have focused primarily on social, economic, and health-system barriers and enablers of RHD care; there is limited evidence from Ethiopia capturing the perspectives of children with RHD (C-RHD) and their caregivers on secondary prophylaxis delivery.

In Ethiopia, access to BPG is commonly supported through a mixed financing structure, including community-based health insurance and out-of-pocket payments. However, operational limitations such as medication stockouts in public facilities may require patients to purchase medicines privately, increasing financial burden. This fee-for-service context may influence adherence patterns and treatment continuity among socioeconomically disadvantaged households and differs from fully subsidized RHD programs implemented in some high-income settings [36,37].

Recent qualitative studies have increasingly emphasised the importance of incorporating child and family perspectives in understanding rheumatic fever and RHD care experiences. Child-centred research has demonstrated that children's experiences of chronic injection-based therapies are shaped not only by clinical factors but also by social, emotional, and relational contexts [38,39]. Studies conducted among Indigenous and Pacific populations in Australia and New Zealand have similarly highlighted the central role of cultural context, family engagement, and health-system relationships in shaping adherence to secondary prophylaxis [40–43]. Integrating insights from these studies strengthens the understanding of lived experience across diverse settings and supports the relevance of exploring children's and caregivers' perspectives within the Ethiopian context.

Thus, the aim of the present study was to explore the factors influencing the utilization of BPG for secondary prophylaxis in C-RHD by examining the perspectives of both C-RHD and their caregivers. Utilizing a qualitative approach, we identified key factors affecting BPG delivery, providing insights to inform healthcare practices and policy decisions to enhance prophylactic treatment delivery and improve outcomes for RHD.

## Methods

### Study design

A qualitative descriptive study [44] with cross-sectional design was conducted, using a semi-structured interview guide for focus group discussion (FGD). To complement prior provider-focused work [45], the present study focused on capturing the lived experiences of C-RHD and their caregivers, specifically to ascertain their perception of BPG secondary prophylaxis, barriers to regular injections, and engagement with available RHD services. The Consolidated Criteria for Reporting Qualitative Research (COREQ) [46] was used to design, conduct and report study findings (S1 Table: COREQ Checklist).

### Study setting

The study was conducted at four major public hospitals in Ethiopia that serve as primary providers of ARF/RHD care in their respective catchment areas: Arba Minch (Arba Minch Comprehensive Specialized Hospital; South Region), Debre Tabor (Debre Tabor Comprehensive Specialized Hospital; Amhara Region), Dire Dawa (Dil-Chora Referral Hospital, Dire Dawa City Administration), and Gondar (University of Gondar Hospital; Amhara Region). These hospitals each serve

millions of residents and are the sole referral or general hospitals in each respective area [47]. Sites were chosen for RHD service availability, geographic spread (northwest, south, east), and accessibility to capture variation in patient experiences.

## Participants

Children aged <18 years of age and living with RHD who had been receiving specialised care and treatment for ARF/RHD for at least six months were recruited via purposive sampling. This population group was specifically selected due to the high burden of disease in children, facilitating a deeper exploration of patient and family perspectives within a relatively homogeneous study cohort [48]. In cases where participants were minors (<14 years), or when their caregivers were available and willing to participate, caregivers were also invited to take part in FGDs to capture caregiving demands, decision-making, and support needs related to BPG prophylaxis. Each FGD was designed to include 5–12 participants. Sample size was determined using principles of thematic saturation [48] for a relatively homogeneous paediatric cohort with narrowly defined objectives. Participants were purposively recruited by the principal researcher (EMB) and the case team leaders for RHD services at the respective hospitals and then scheduled for the FGDs.

Participants' ethnicity and primary language were not collected in the sociodemographic survey. This was a deliberate decision based on cultural sensitivity and the prevailing sociopolitical context at the time of data collection, including heightened ethnic-based tensions, such that requesting ethnic identification was considered inappropriate and could have undermined trust and participation. All child participants were Ethiopian nationals, and each study site served populations that are largely linguistically and culturally homogeneous within their respective regions.

## Data collection

Discussion topic guides were developed and initially piloted with three C-RHD prior to FGDs to enhance question clarity and validity. Minor modifications were implemented to improve flow and pacing of the discussions. Standardized discussion guides were designed to assess factors impacting BPG delivery across structural, organizational, therapeutic, provider-, and individual-level levels (S1 Text: *Discussion guides*). The guide comprised open-ended prompts constructed to elucidate participants' experiences with ARF/RHD services, perceived barriers/facilitators to BPG delivery, treatment experiences (including pain and injection logistics), and healthcare accessibility. Participants were invited to offer insights on potential improvements to secondary prophylaxis services for RHD.

Focus group discussions were conducted between 20 May and 29 July 2022, with one session held at each site (four in total). Written caregiver consent and child assent were obtained prior to participation. FGDs were facilitated by a trained investigator (EMB; male; Ethiopian national), with the assistance of an independent male postgraduate research student at one site. Discussions were conducted in Amharic at three sites which was a shared working language between participants and facilitators. At Dire Dawa, FGDs were conducted entirely in Afan Oromo. In all sites, facilitators were fluent in the language used and no external interpreters were involved. To minimise potential power imbalances during FGDs involving children and caregivers, facilitators actively encouraged equitable participation and ensured that children were invited to share their perspectives directly. Caregivers were reminded to allow children to express their views independently where appropriate. Facilitators used age-appropriate language and prompts to support engagement of younger participants and maintain a child-centred discussion environment. Sessions lasted 30–60 minutes, were audio-recorded, and participants were assigned codes to support confidentiality and discussion flow. Audio recordings were translated into English for analysis by a bilingual member of the research team (EMB), with independent verification to ensure accuracy and consistency. Although specialised child-centred tools such as role-play or visual methods were not employed, semi-structured group discussions were considered appropriate given the age distribution of participants, most of whom were adolescents capable of articulating experiences verbally. This approach aligns with established qualitative studies

involving children with chronic conditions, where structured discussion formats have been successfully used to capture experiential data [38–40,43].

## Data analysis

Audio recordings were transcribed verbatim, translated into English, and analysed using a framework approach. Data were managed in QSR NVivo (Release 1.7.1, QSR international, Massachusetts, USA) for data organization and analysis.

Inductive open coding was used to develop and iteratively refine a codebook across the full dataset. Codes were clustered into themes and organised using Chaudoir et al. [49], implementation domains (structural, organisational, therapeutic/product, provider, and patient/individual). To interpret behavioural mechanisms relevant to children and caregivers, themes were then mapped to the COM-B model (S2 Table) as shown in Fig 2. Mapping themes to the COM-B model was undertaken to interpret behavioural, relational, and structural determinants of treatment engagement. In this study, COM-B was applied as an implementation framework to identify influences on capability, opportunity, and motivation at individual, interpersonal, and health-system levels, rather than to attribute responsibility to participants [50]. Two investigators (EMB, BRM) independently coded the transcripts; discrepancies were resolved by discussion until consensus, with an audit trail maintained. This strategy linked lived experiences to implementation determinants and behaviour-change levers for improving BPG secondary prophylaxis.

## Ethical approval

Ethics approval of the study was granted by the Institutional Review Board at the University of Gondar (VP/RTT/05/756/2022) and the Human Research Ethics Committee at Curtin University (HRE2022−0221). Formal written consent was obtained from caregivers, and assent was appropriately secured from C-RHD.

## Results

A total of 42 participants (C-RHD: n = 30; caregivers: n = 12) were recruited from four hospitals: Arba Minch (n = 12), Debre Tabor (n = 10), Dire Dawa (n = 8), and Gondar (n = 12). The mean age of C-RHD participants was 13.5 ± 2 years, and caregivers (all were parents except one adult sibling) had a mean age of 36.7 ± 8 years. Among the 18 children under 14 years of age, six had caregivers who were unable to participate due to scheduling conflicts.

While eligibility permitted inclusion of children with ARF or RHD receiving ≥6 months of secondary prophylaxis, all enrolled participants had confirmed symptomatic RHD at recruitment; no cases of isolated ARF were represented in the final cohort. All participants living with RHD were receiving ongoing clinical follow-up at their respective centres. Fourteen (47%) had symptomatic RHD without heart failure, while 16 (53%) had RHD complicated by heart failure; none had advanced or surgically treated disease (e.g., prior valve surgery or end-stage decompensation). Three participants had prior exposure to BPG but were subsequently switched to oral amoxicillin, and five were initiated on amoxicillin, primarily due to healthcare provider reluctance to administer injections and partially related to drug stockouts. Key sociodemographic and clinical characteristics are summarized in Table 1 (presented as frequencies, percentages, or means ± SD, as appropriate).

Themes were organised using Chaudoir et al's [51] implementation domains (structural, organisational, therapeutic/product, provider, and patient/individual, (Fig 1) and then mapped them to COM-B model to interpret determinants of treatment engagement (Fig 2). To foreground the child/family perspective, results prioritise (i) children's psychological distress and fear of injections, (ii) caregivers' travel/financial burden, and (iii) mistrust/confusion when BPG is substituted with oral antibiotics. A word cloud summarising salient terms is provided (S1 Fig: *Word cloud*).

## Structural factors

**Service inaccessibility and travel/financial burden.** Geographic distance and transportation costs were consistently identified as major structural barriers to maintaining regular secondary prophylaxis. Participants described the cumulative

**Table 1. Sociodemographic and clinical data of C-RHD participants.**

| Variables | | Site | | | | Total |
|---|---|---|---|---|---|---|
| | | Arba Minch | Dire Dawa | Debre Tabor | Gondar | |
| Age, years (mean±SD) | | 11.8±3 | 12.5±2 | 13.9±2 | 15.0±2 | 13.5±2 |
| Gender | Male | 3 | 2 | 3 | 5 | 13 (43%) |
| | Female | 5 | 4 | 5 | 3 | 17 (57%) |
| Residency | Rural | 3 | 4 | 5 | 4 | 16 (53%) |
| | Urban | 5 | 2 | 3 | 4 | 14 (47%) |
| Comorbidity | Yes | 1 | 0 | 0 | 0 | 1 (3%) ‡ |
| | No | 7 | 8 | 6 | 8 | 29 (97%) |
| Concomitant drug | Yes * | 2 | 2 | 5 | 7 | 16 (53%) |
| | No | 6 | 4 | 3 | 1 | 14 (47%) |
| Severity of RHD | Symptomatic without heart failure | 2 | 2 | 5 | 7 | 16 (53%) |
| | Symptomatic with heart failure | 6 | 4 | 3 | 1 | 14 (47%) |
| Perceived improvement in disease progress † | Improved | 5 | 4 | 6 | 5 | 20 (67%) |
| | No improvement | 3 | 2 | 2 | 3 | 10 (33%) ** |
| Secondary prophylactic agent | BPG | 8 | 6 | 8 | 0 | 22 (73%) |
| | Amoxicillin | 0 | 0 | 0 | 8 § | 8 (27%) |

†Perceived C-RHD caregiver self-reported, ‡ patients with heart failure secondary to ARF/RHD are excluded, * The concomitant drugs were all related to RHD treatment (diuretics, angiotensin converting enzyme inhibitors), ** Includes C-RHD who said the change is mild and/or exacerbated. § three of them were having previous BPG use experience, however, required to switch to amoxicillin after healthcare providers resistance to administer BPG.

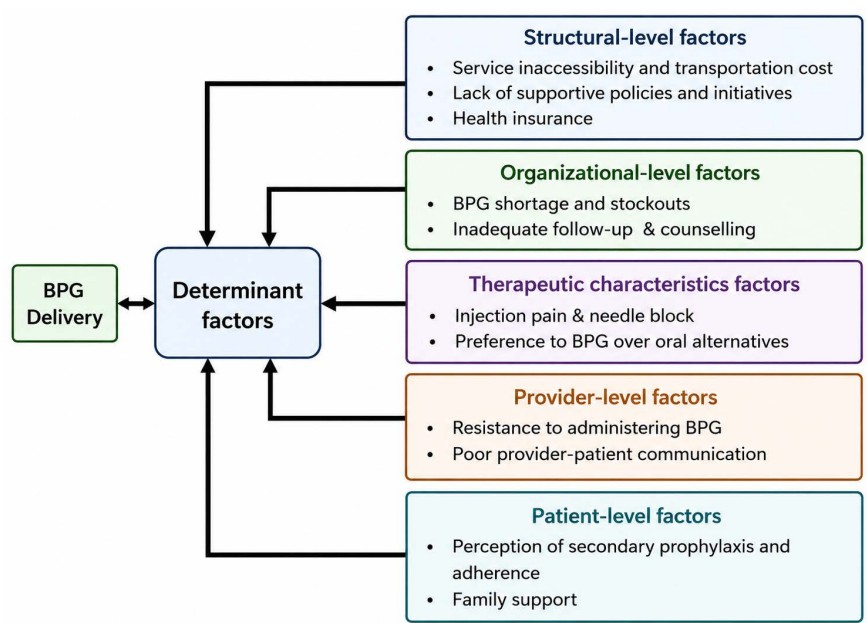

**Fig 1. Determinants of BPG delivery from children/caregivers' perspectives, organised by Chaudoir domains (adapted from Chaudoir et al. [51]).**

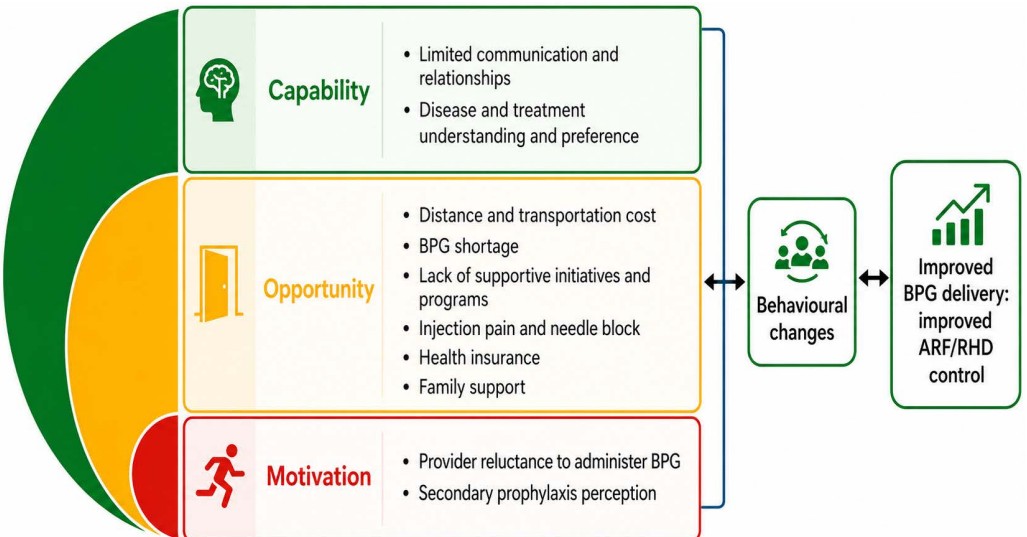

**Fig 2. Themes mapped to COM-B (capability, opportunity, motivation) emphasising determinants of treatment engagement.**

financial burden associated with repeated travel to centralized hospitals, particularly among rural households where transport expenses represented a substantial proportion of monthly income. These constraints often resulted in delayed clinic attendance or missed injections, demonstrating how physical access limitations directly influenced treatment continuity.

*"… however, there are times when transportation is an issue since we come from a distant location, and we sometimes pay up to 400 Ethiopian birr for contracted transport services"*

*C-RHD, Dire Dawa*

**Limited supportive policies and inconsistent financial protections.** Participants highlighted the absence of dedicated financial support programs for RHD care, frequently comparing their situation with other public health services that were provided free of charge. This perceived lack of prioritization contributed to ongoing financial strain and uncertainty regarding treatment continuity. Many participants advocated for government-supported or subsidized provision of BPG to reduce the economic burden associated with long-term treatment.

*"Another thing very important is that I don't think it will be beyond the capacity of the government and the public to support RHD treatment. For example, malaria treatment and vaccinations are now given free of charge. It would be good if the government knew that we have had this disease for a long time and that we have economic problems to arrange support".*

*Caregiver, Arba Minch*

**Community-based health insurance: Partial relief, limited impact during stockouts.** Community-based health insurance schemes were reported to reduce direct out-of-pocket costs for some families. However, participants noted that these benefits were limited when medications were unavailable at public facilities, forcing them to purchase BPG from private pharmacies at substantially higher costs. This inconsistency reduced the protective effect of insurance and created

ongoing financial uncertainty. Participants described the dual benefits and limitations of insurance. Most participants who noted the importance of health insurance were from rural areas.

*"In the past, there was no health insurance, so the cost was unaffordable, but now, because there is health insurance, it is good. We don't have difficulty in adhering the follow-up or the medicine"*

*C-RHD, Gondar*

### Organizational factors

**BPG shortage and stockouts.** A recurring theme among most participants, from both rural and urban areas across all study sites, was the challenge of accessing medication. Many noted that *"the drug is sometimes unavailable at the hospital and is costly at private pharmacies".* Consequently, some individuals who could not obtain BPG *"… switched to amoxicillin".*

*"The drug is sometimes unavailable, and private pharmacies charge high prices. Many have switched to amoxicillin".*

*Caregiver, Gondar*

Families described coping strategies (e.g., purchasing extra vials when available) reflecting chronic supply uncertainty and they recommended the establishment of reliable supply chain to prevent BPG stockouts and *"make getting BPG simple".*

*"Yes, we usually come by appointment and continue with follow-ups. However, since the medicine is often unavailable here, we need to purchase it from outside sources. I plan to buy and store it for the next month because we're concerned about its availability".*

*Caregiver, Arba Minch*

**Rushed follow-up and limited counselling.** Participants frequently described clinic visits as brief and task-focused, with limited opportunity for discussion about disease progression, treatment expectations, or long-term management. This perceived lack of engagement reduced confidence in care and limited understanding of treatment goals. In addition, poor record-keeping systems were reported to interrupt continuity of care and contributed to treatment discontinuation in some cases.

Participants described these service limitations:

*"Currently, doctors only prescribe medication each month without providing thorough follow-up care. Addressing this issue could offer us better hope for improvements"*

*C-RHD, Debre Tabor*

### Therapeutic/product characteristics

**Severe injection pain and needle blockage drive fear and distress.** Injection-related pain emerged as a central experiential barrier, not necessarily leading to missed doses but contributing to anticipatory anxiety and emotional distress. Participants frequently described procedural difficulties such as needle blockage, reinforcing the perception that the treatment process itself required improvement.

*"Although the medication itself has been effective, the injection process is painful. Sometimes, if the needle gets blocked during the injection, it takes two or three additional attempts, which can be frustrating"*

*C-RHD, Debre Tabor*

Despite distress, most prioritised adherence given perceived efficacy, and advocated for pain-management strategies. **Mistrust and confusion when BPG is substituted with oral antibiotics.** Participants described variations in treatment practices across study sites, particularly regarding substitution of BPG with oral antibiotics. In several locations, BPG was consistently administered, whereas in Gondar, participants reported that injections were sometimes withheld and replaced with oral amoxicillin. These changes created confusion among families and reduced confidence in treatment decisions, particularly when explanations for substitution were limited. Some participants perceived worsening symptoms following changes in treatment, which reinforced their belief in the effectiveness of injectable therapy and increased concern about treatment substitution.

Participants described their experiences with treatment changes:

*"I used to take injections, and they were more effective. The injections made a noticeable difference, but after switching to oral amoxicillin, my condition worsened…. When I started taking amoxicillin, I experienced chest pain. Previously, I could walk without difficulty, but now walking has become challenging. Although the injection caused immediate spreading pain, it was still effective and had better results"*

*C-RHD, Gondar*

However, a minority of participants expressed preference for oral medications due to concerns about injection-related discomfort or mobility limitations. These differing preferences highlight the need for clear communication and individualized counselling when treatment modifications are considered.

*"Maybe if there is an oral drug instead because sometimes it makes me unable to walk. I know that if this injection goes on, it will have a negative impact in the future. So, I would say if the medicine were replaced by an oral one"*

*C-RHD, Debre Tabor*

### Provider-level factors

**Reluctance to administer BPG due to fear of severe reactions.** Provider reluctance to administer BPG emerged as a significant barrier to treatment continuity, particularly in settings where concerns about severe reactions, including anaphylaxis, influenced clinical decision-making. Participants described instances in which fear among healthcare providers led to refusal to administer injections, resulting in delays or substitution with oral prophylaxis; this was most pronounced in Gondar. This reluctance contributed to fragmented care pathways, requiring families to seek alternative service providers to maintain treatment.

Caregivers described the challenges associated with provider hesitancy:

*"… Everyone kept saying, 'I am afraid,' while I was left wondering if they had any compassion for my situation. The patient was in their care, not mine. I asked them if I should just take him home without the injection"*

*Caregiver, Arba Minch*

Participants also highlighted the consequences of service fragmentation, including the need to seek alternative facilities capable of administering injections:.

*"[The hospital] no longer administers injections. As a result, I had to take the medication to local health centres where they were able to give the injections. They claim that administering the injections is difficult and painful, and they are unwilling to do it".*

*Caregiver, Gondar*

Many participants emphasised the importance of having trained and dedicated personnel available to administer injections safely and consistently, suggesting that improved provider training and confidence could strengthen trust in service delivery.

**Limited communication undermines motivation and adherence.** Limited communication between healthcare providers and families emerged as an important barrier affecting treatment understanding and engagement. Participants frequently described clinic encounters that focused primarily on medication administration rather than discussion of treatment goals, disease progression, or long-term management. This lack of counselling reduced confidence in care and contributed to uncertainty regarding treatment expectations, potentially undermining motivation to maintain regular prophylaxis.

Participants expressed the need for more consistent counselling and education:

*"It would be beneficial if we received more comprehensive support, such as counselling services, since they are here to treat us and share their expertise. However, we have not been given any education on these matters"*

*Caregiver, Arba Minch*

### Individual (Patient)-level factors

**Treatment fatigue and uncertainty about duration.** While most participants recognised the importance of regular BPG injections in preventing disease progression, uncertainty regarding the duration of treatment emerged as an important source of psychological burden. Participants frequently expressed concerns about the long-term commitment required for monthly injections, particularly when treatment duration was not clearly explained. This uncertainty contributed to treatment fatigue and reduced motivation among some families, highlighting the importance of consistent counselling regarding treatment timelines and expected outcomes.

Participants described uncertainty about treatment duration:

*"I don't have a specific opinion on the medicine itself, but I am concerned about the duration of the treatment. My daughter keeps asking how long she will need to continue the injections"*

*Caregiver, Arba Minch*

Interruptions to treatment, often resulting from medication shortages or provider reluctance, were perceived to have direct negative effects on symptom control. Participants described symptom worsening during periods of missed injections, reinforcing the perceived importance of maintaining uninterrupted prophylaxis. These experiences strengthened participants' recognition of the need for reliable medication supply and consistent provider support.

Participants reflected on the consequences of treatment interruption:

*"I have previously begun the treatment and then stopped… and during those periods of interruption, my pain worsened… my main concern is the need to continuously [take] prophylactic medication as a palliative measure"*

*C-RHD, Debre Tabor*

**Family support.** Family support emerged as a critical facilitator of treatment continuity, particularly in the context of financial and logistical challenges. Many participants described reliance on caregivers and family members to coordinate travel, provide emotional encouragement, and maintain adherence to scheduled clinic visits. This collective support was often described as essential for sustaining long-term engagement with treatment despite ongoing hardships.

Participants emphasised the importance of family involvement:

*"It's only with my family's support that I can come here, and they have been very understanding throughout this process"*

*C-RHD, Debre Tabor*

## Discussion

This study examined barriers and facilitators to BPG uptake from the perspectives of C-RHD and their caregivers, applying an implementation framework [49] and COM-B model [50]. Beyond confirming well-described supply and service barriers, our analysis highlights determinants that were particularly salient to families psychological distress related to injections, travel and financial strain, mistrust when BPG is substituted with oral antibiotics, and reliance on family support networks thereby extending evidence derived from provider-focused work [45].

Our findings align with emerging qualitative research from Indigenous and Pacific populations, where structural factors such as service accessibility, health system responsiveness, and cultural trust have been identified as central determinants of adherence to secondary prophylaxis [40–43]. Many of the most influential determinants were structural and relational, including transport burden, BPG stockouts, out-of-pocket costs, provider fear of severe reactions, fragmented referral pathways, and limited counselling. These determinants should not be interpreted as individual failings of children or caregivers; rather, they reflect the interplay between resources, health-service organisation, provider confidence, communication quality, and medicine availability [52,53]. This interpretation aligns with contemporary lived-experience and decolonising RHD literature, which emphasises structural inequity, stigma, and the importance of centring patient and family perspectives in care design [54]. Collectively, these findings highlight that many barriers to sustained BPG delivery arise from systemic and organisational constraints, directing attention toward transport access, supply continuity, service organisation, and therapeutic relationships rather than individual-level motivation [50].

Our findings both align with and extend prior qualitative work conducted in Uganda [26], which explored the lived experiences of people with RHD across a broad age range using in-depth individual interviews and identified social, economic, and health-system barriers to care. While similar structural barriers, particularly transport costs, drug availability, and dependence on family support were observed in both settings, important methodological and population differences explain the distinct insights generated. The Ugandan study included adolescents and adults with varying disease severity and examined overall RHD care experiences, whereas our study used focus group discussions restricted to C-RHD and their caregivers. The design of the present study enabled exploration of shared child–caregiver dynamics, caregiver-mediated decision-making, centre-level variation in BPG delivery, and therapeutic challenges specific to secondary prophylaxis, including injection pain, needle blockage, provider refusal to administer BPG, and consequences of switching to oral regimens. Together, these findings suggest that although structural barriers are consistent across endemic settings, child-specific distress, caregiver burden, and provider risk perception may play a more central role in shaping BPG delivery.

A central discrepancy emerged between what families need (reassurance, consistent access, pain management, counselling, and financial protection) and what providers fear (anaphylaxis and liability), with the latter contributing to refusal to inject BPG in one centre and consequent switches to less effective oral regimens. This misalignment risks eroding trust and adherence despite strong family motivation to persist with BPG. At system level, these findings underscore the

importance of embedding RHD care within organised, family-centred programmes and registries [55,56], supported by national strategies [57,58] and integration into primary care and universal health coverage pathways.

Structurally, families described distance and transport costs as recurrent barriers, sometimes approximating a day's wage [59], with centralised services amplifying burden for rural households [60]. Community-based health insurance offered partial relief yet proved insufficient during stockouts [24,25,61]. These family-level accounts stress that financial and geographic barriers are not abstract system issues but lived constraints shaping monthly decisions to attend for injections.

Consistent with World Health Organization (WHO) recommendations [13,62] and Ethiopian practice [33], families strongly preferred BPG; however, intermittent stockouts and healthcare providers' refusal to inject resulted in missed doses or substitution with oral agents perceived as less effective [63–65]. Reports of rushed follow-ups and limited counselling further undermined confidence, likely reflecting system overload rather than lack of provider commitment.

Children frequently described severe injection pain and needle blockage, fuelling anticipatory anxiety [26,66–68]. Notably, pain was rarely a cause of missed doses; instead, families requested practical supports (e.g., analgesia) to sustain adherence [68,69]. Where substitution to oral therapy occurred, participants commonly perceived clinical deterioration, reinforcing the patients' preference of maintaining BPG regimens, consistent with therapeutic guidelines [16,63,69].

Provider-level reluctance to inject driven by fear of severe reactions and medicolegal repercussions emerged as a decisive barrier in some settings, precipitating switches to oral regimens [16,45,70]. This aligns with, but is more pronounced than, reports from other low-resource contexts [24,25], suggesting a need for behaviour-change interventions that address risk perception, clarify roles/liability, and ensure readiness to manage adverse events.

Although severe adverse events are uncommon [71], evidence indicates that most serious reactions following BPG administration are non-allergic in nature, including vasovagal events, inadvertent intravascular injection, and cardiovascular collapse rather than true IgE-mediated anaphylaxis [72]. Fatal events are rare and are more frequently non-allergic than allergic. Recent WHO guidance [62] and American Heart Association reports [70] emphasise that fear of anaphylaxis should not preclude BPG administration when delivered using correct intramuscular technique by trained personnel with appropriate readiness to recognise and manage acute reactions. These recommendations prioritise competency-based training, standardised protocols, post-injection observation, and availability of basic emergency equipment rather than avoidance of BPG. In this context, provider fear, particularly in the absence of structured training and institutional guidance may inadvertently drive substitution with less effective oral regimens, potentially increasing the long-term risk of ARF recurrence and RHD progression.

Emerging evidence further suggests that alternative delivery routes may help address injection-related barriers identified in this study. Recent population pharmacokinetic studies demonstrated that subcutaneous administration of BPG achieved sustained and predictable penicillin concentrations, with favourable pharmacokinetic characteristics for secondary prevention of RHD [73,74]. Subcutaneous delivery may reduce injection-related pain and procedural challenges, potentially improving acceptability for C-RHD [75] and confidence among healthcare providers. However, evidence regarding safety, tolerability, and programmatic feasibility in paediatric and low-resource settings remains limited, and further clinical and implementation research is required before routine adoption of subcutaneous delivery BPG.

Strengthening communication and patient/carer counselling, including duration expectations and pain-management plans, may counter treatment fatigue and sustain motivation, while improved formulations and delivery strategies could bolster confidence for both providers and families. Individually, children and caregivers balanced perceived benefits against treatment fatigue and uncertainty about duration. Regular, empathetic counselling and family-centred education may help maintain adherence over years of prophylaxis [76].

## Limitations

Findings reflect a modest sample drawn from four hospitals and the experiences of paediatric patients and families, which may limit transferability to adults or non-attenders. Translation may introduce nuance loss despite quality checks.

Site-level practice differences in clinical practice (e.g., substitution of therapeutic agents) may also shape participant perceptions. In addition, children and adolescents were analysed as a single cohort, which may have obscured developmental differences in perceptions of treatment. While efforts were made to encourage equitable participation, the inclusion of caregivers in group discussions may have influenced the extent to which younger participants expressed independent views. Ethnicity data were not collected due to cultural and sociopolitical sensitivities, limiting exploration of potential contextual influences. Nonetheless, triangulation through independent coding and application of a structured implementation framework strengthen the credibility and analytic rigour of the study.

### Implications

Our child/family-centred results complement provider-focused evidence [45] to offer a more comprehensive view of BPG delivery. Priorities include (i) reliable BPG supply and decentralised access; (ii) provider training and team-based protocols for anaphylaxis readiness to reduce fear and clarify roles; (iii) routine local anaesthetics and practical pain-reduction strategies; (iv) structured counselling addressing duration, goals, and options when stockouts occur; and (v) financial and logistical supports (e.g., transport assistance) to reduce household burden. Integrating these elements within register-based programmes can align provider safety concerns with families' needs, sustaining adherence and improving outcomes.

### Conclusion

Guideline-concordant delivery of BPG is critical to controlling ARF/RHD after diagnosis. From the perspectives of children and caregivers, uptake is constrained by recurrent BPG shortages, severe injection pain and needle blockage, provider reluctance to inject, and rushed, centralized services. Tackling these barriers requires dependable BPG supply and decentralized access; routine analgesia and practical pain-reduction measures; team training and anaphylaxis-readiness to reduce provider anxiety and restore confidence in BPG use; structured counselling on duration and goals; and financial/transport supports for families. Embedding these elements within register-based, family-centred programs and primary-care/universal health coverage pathways in Ethiopia may improve continuty of secondary prophylaxis and support better long-term outcomes for C-RHD.

### Supporting information

**S1 Text. Discussion guides.**
(DOCX)

**S1 Table. COREQ checklist.**
(DOCX)

**S2 Table. COM-B model domains definition.**
(DOCX)

**S1 Fig. Word cloud of transcripts.**
(DOCX)

### Acknowledgments

We extend our gratitude to all study participants for their time and willingness to share their experiences and perceptions. We also thank Delelegn Getachew for assisting with the FGDs at Dil-Chora Hospital in Dire Dawa. Additionally, we acknowledge Curtin University for supporting Eshetie Melese Birru, the recipient of the Curtin University International Postgraduate Research Scholarship.

## Author contributions

**Conceptualization:** Eshetie Melese Birru, Kevin T. Batty, Brioni R. Moore.

**Data curation:** Eshetie Melese Birru, Stephanie L. Enkel, Brioni R. Moore.

**Formal analysis:** Eshetie Melese Birru, Brioni R. Moore.

**Investigation:** Eshetie Melese Birru, Kevin T. Batty, Laurens Manning, Brioni R. Moore.

**Methodology:** Eshetie Melese Birru, Kevin T. Batty, Brioni R. Moore.

**Project administration:** Eshetie Melese Birru.

**Resources:** Eshetie Melese Birru, Kevin T. Batty.

**Software:** Eshetie Melese Birru.

**Supervision:** Eshetie Melese Birru, Kevin T. Batty, Laurens Manning, Brioni R. Moore.

**Validation:** Eshetie Melese Birru, Kevin T. Batty, Stephanie L. Enkel, Brioni R. Moore.

**Visualization:** Eshetie Melese Birru, Kevin T. Batty, Stephanie L. Enkel, Brioni R. Moore.

**Writing – original draft:** Eshetie Melese Birru.

**Writing – review & editing:** Eshetie Melese Birru, Kevin T. Batty, Laurens Manning, Stephanie L. Enkel, Brioni R. Moore.

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
