## [Decision Letter · Decision Letter 0]

5 Feb 2026

PONE-D-25-52910Determinants of benzathine penicillin G delivery for childhood rheumatic heart disease in Ethiopia: a qualitative study of children and caregiversPLOS One

Dear Dr. Birru,

Thank you for submitting your manuscript to PLOS ONE. After careful consideration, we feel that it has merit but does not fully meet PLOS ONE’s publication criteria as it currently stands. Therefore, we invite you to submit a revised version of the manuscript that addresses the points raised during the review process.

We look forward to receiving your revised manuscript.

Kind regards,

Emma Campbell, Ph.D

Staff Editor

PLOS One

Journal Requirements:

4. We are unable to open your Supporting Information file “Supplementary Text file 1_Discussion guides.docx and Supplementary Table 2-COM-B Model domain definitions.docx” Please kindly revise as necessary and re-upload.

Additional Editor Comments:

The manuscript has been assessed by two reviewers and their comments are available below. They have provided some constructive feedback and requested further clarity around some of the methodology. Please review their comments below and make the appropriate revisions. Please note that any suggestions for additional references should be fully reviewed to ensure they are appropriate before they are included, there is no obligation to include the suggested references.

Reviewer's Responses to Questions

**Comments to the Author**

1. Is the manuscript technically sound, and do the data support the conclusions?

Reviewer #1: Yes

Reviewer #2: Yes

2. Has the statistical analysis been performed appropriately and rigorously? 

Reviewer #1: Yes

Reviewer #2: N/A

3. Have the authors made all data underlying the findings in their manuscript fully available?

Reviewer #1: Yes

Reviewer #2: No

4. Is the manuscript presented in an intelligible fashion and written in standard English?

Reviewer #1: Yes

Reviewer #2: Yes

5. Review Comments to the Author

Reviewer #1: This is an intersting study , reported highly needed information about RHD Control

1. The title: suggest to change to Determinants of " secondary prophylaxis for RHD" delivery as the study contains 30% of patients not recieving BPG

2. Introduction: need to emphasize the diffences between this study and the similar one that was conducted in Uganda

3. Methods: the patient information regarding clinical status is important (stage of disease, heart failure) , also need to include the reasonn for changing form injection to oral as that affects many other factors

4. Discussion: more in depth discussion about BPG side effects, they include nonallergic fatalities and the emphasis on allergy needs to be less, rather , discuss this nonallergic reactions and need for training the health care personnel following WHO latest guidelines 2024 and reports from the American H Association

https://www.ahajournals.org/doi/10.1161/JAHA.121.024517

5. Comparing the results of this study with the similar one done in Uganda is needed emphasizing methodologic differences and outcomes

6. May include in discussion the research on subcutaneous BPG

Kado JH, Salman S, Henderson R, Hand R, Wyber R, Page-Sharp M, Batty K, Carapetis J, Manning L. Subcutaneous administration of benzathine benzylpenicillin G has favourable pharmacokinetic characteristics for the prevention of rheumatic heart disease compared with intramuscular injection: a randomized, crossover, population pharmacokinetic study in healthy adult volunteers. J Antimicrob Chemother. 2020 Oct 1;75(10):2951-2959. doi: 10.1093/jac/dkaa282. PMID: 32696033.

Reviewer #2: General assessment

This paper reports a qualitative study on barriers and facilitators to use of benzathine penicillin prophylaxis for children with rheumatic heart disease in Ethiopia. The study has considerable value in providing a child and family perspective to complement provider-based research. Overall the paper is written clearly, uses appropriate methodology and is structured well. Quoted excerpts from comments provided by participants in focus group greatly enhance the results, and the discussion section contextualises findings with reference to the wider literature and implementation frameworks. There are a small number of areas that could be improved, and I recommend acceptance with minor revisions to address these areas.

Specific comments

Methods

Participant inclusion

Child participants in the study were defined as "C-RHD <18 years of age who were receiving specialised care and treatment for ARF/RHD for at least 6 months..." (Methods paragraph 4, page 4). The term "ARF/RHD" here creates ambiguity here: did all the children have confirmed RHD but some had a history of diagnosed ARF, or does it mean that children could be included if they had either ARF or RHD (or both), therefore some might have had ARF alone and no RHD? If the latter is true, the descriptor "C-RHD" (children with RHD) seems inaccurate, and the lived experience of those who were solely diagnosed with ARF is likely different from those diagnosed with RHD. I suggest further clarification, for example that all child participants had been diagnosed with RHD, if this is correct.

Researcher positionality and language

Positionality is only described in terms of gender (paragraph 2, page 5), and ethnicity, nationality and language fluency are not mentioned. The paper does not clearly state the language used in the focus group discussions, or whether multiple languages were used. Transparency in this area is important to allow readers to consider possible bias, translation effects and dynamics between participants and facilitator/researchers. I suggest that the authors briefly expand on the information provided about the focus group methods, to include ethnicity of facilitators, language(s) spoken during focus groups, and whether researchers shared language(s) with participants or whether translators were used during the focus groups.

Results

Participants' ethnicity and language

Sociodemographic characteristics provided on children with RHD participating in the study (Results paragraph 1, page 5, and Table 1, pages 5 and 6) do not include ethnicity or primary languages spoken. It would be helpful to include these data if available to help readers' understanding of the diversity of participants.

Participant clinical characteristics

The clinical characteristics of child participants presented in Table 1, pages 5 and 6, does not provide information on whether RHD was present (see comment above referring to the use of the term "ARF/RHD" in inclusion criteria). On the assumption that all child participants did have diagnosed RHD, it would be useful to provide further subcategorisation of the severity of illness, for example whether their illnesses were symptomatic or asymptomatic, if these data are available. This would help provide further context for interpreting the results, as the lived experience of RHD of those with latent disease would likely differ from those with overt illness.

Discussion

Provider refusal to inject

In Discussion paragraph 2 (page 11) it is indicated that there was "refusal to inject in some centres". According to the results this appeared to be occurring only in one centre. I suggest that this is reworded to "...refusal to inject in one centre..."

Supplementary 1 Text

Wording of questions/prompts used in focus group discussions

As described in the Methods (paragraph 1, page 5), Supplementary 1 Text provides the guide for facilitating the focus group discussions, and contains the open-ended prompts used in these discussions. The prompt provided for factors impacting prophylaxis delivery services ("What factors influence the delivery of BPG prophylaxis in healthcare settings?") uses health service wording that would not be clear to child participants with a mean age of 13.5 years. It would be helpful to provide the prompt as actually used in the focus group discussions, translated into English.

This manuscript makes a valuable contribution by centring the perspectives of children and caregivers in RHD prophylaxis provision. The areas I have identified here, where minor clarification would aid readers, do not detract from the overall quality of the work and suitability for publication.

6. PLOS authors have the option to publish the peer review history of their article (what does this mean?). If published, this will include your full peer review and any attached files.

Reviewer #1: **Yes:** Sulafa KM Ali

Reviewer #2: No

---

## [Author Response · Author response to Decision Letter 1]

19 Feb 2026

Dear Editor,

Thank you for the opportunity to submit a revised version of the article entitled “Determinants of secondary prophylaxis for childhood rheumatic heart disease in Ethiopia: a qualitative study of children and caregivers". We are grateful to the reviewers for the opportunity to improve this piece of work and have carefully addressed each of their comments and revised the manuscript accordingly. In particular, we have clarified participant inclusion criteria, expanded reporting of clinical characteristics, strengthened the Discussion with relevant comparative and contextual literature, and improved methodological transparency where suggested.

A detailed, point-by-point response to each comment is provided below, with corresponding revisions clearly indicated in the manuscript.

1. (R1): The title: suggest to change to Determinants of " secondary prophylaxis for RHD" delivery as the study contains 30% of patients not receiving BPG.

We agree and have revised ((Page 1, Line 1-2).

2. (R1): Introduction: need to emphasize the differences between this study and the similar one that was conducted in Uganda

We thank the reviewer for this important comment. We have revised the Introduction to more clearly distinguish the present study from the similar qualitative study conducted in Uganda, and have added an encompassing reference to the social, economic, and health-system barriers and enablers to RHD care assessed across these studies (Page 4, Line 99-102).

3. (R1): The patient information regarding clinical status is important (stage of disease, heart failure), also need to include the reason for changing from injection to oral as that affects many other factors.

We agree that clinical status is important for interpreting participants’ experiences and factors influencing secondary prophylaxis delivery. Although eligibility included ARF and RHD patients on secondary prophylaxis for ≥6 months, all enrolled participants had symptomatic RHD—likely reflecting delayed presentation and diagnosis in resource-limited settings. Of the cohort, 47% had symptomatic RHD without heart failure and 53% had RHD with heart failure; none had complicated RHD. We have clarified these clinical characteristics in the Results section. In addition, we have clearly described the reasons for switching from injectable benzathine penicillin G to oral prophylaxis and these considerations have now been addressed in Results, and Discussion sections (Page 7, Line 225-233).

4. (R1): Discussion: more in-depth discussion about BPG side effects, they include nonallergic fatalities and the emphasis on allergy needs to be less, rather, discuss this nonallergic reactions and need for training the health care personnel following WHO latest guidelines 2024 and reports from the American Heart Association

We thank the reviewer for this important clarification. We agree that an overemphasis on allergic reactions may obscure the broader safety profile of BPG. We have revised the Discussion to (i) de-emphasise allergy as the sole safety concern, (ii) explicitly discuss rare but serious non-allergic adverse events (including cardiovascular collapse and injection-related fatalities), and (iii) emphasise the need for healthcare worker training, preparedness, and adherence to current WHO (2024) guidance and American Heart Association recommendations. These revisions better contextualise provider fear, clarify that most severe events are non-allergic, and highlight actionable system-level solutions rather than avoidance of BPG use (Page 15-16, Line 477-489).

5. (R1): Comparing the results of this study with the similar one done in Uganda is needed emphasizing methodologic differences and outcomes

We thank the reviewer for this valuable suggestion. We agree that clearer comparison with the similar qualitative study in Uganda strengthens interpretation of our findings and have expanded the Discussion accordingly. The Ugandan study used in-depth interviews across a wide age range and examined the broader lived experience of RHD and multilevel barriers and enablers to care. In contrast, our focus groups with children with symptomatic RHD and their caregivers enabled exploration of shared and divergent child–caregiver perspectives and caregiver-mediated decision-making, with a specific focus on BPG delivery (including centre-level variation, provider refusal, practical challenges, and switching to oral prophylaxis). While both studies identified common system constraints, our findings extend the Ugandan work by foregrounding child distress, caregiver burden, and provider risk perception as key influences on BPG delivery in Ethiopia (Page 14, Line 428-441).

6. (R1): May include in discussion the research on subcutaneous BPG. Kado JH, Salman S, Henderson R, Hand R, Wyber R, Page-Sharp M, Batty K, Carapetis J, Manning L. Subcutaneous administration of benzathine benzylpenicillin G has favourable pharmacokinetic characteristics for the prevention of rheumatic heart disease compared with intramuscular injection: a randomized, crossover, population pharmacokinetic study in healthy adult volunteers. J Antimicrob Chemother. 2020 Oct 1;75(10):2951-2959. doi: 10.1093/jac/dkaa282. PMID: 32696033.

As suggested, we have incorporated the above reference and expanded the Discussion to include evidence on subcutaneous administration of benzathine penicillin G (BPG), highlighting its favourable pharmacokinetic characteristics and potential relevance for addressing injection-related barriers identified in this study (Page 16, Line 491-499).

7. (R2): Methods, Participant inclusion: Child participants in the study were defined as "C-RHD <18 years of age who were receiving specialised care and treatment for ARF/RHD for at least 6 months..." (Methods paragraph 4, page 4). The term "ARF/RHD" here creates ambiguity here: did all the children have confirmed RHD but some had a history of diagnosed ARF, or does it mean that children could be included if they had either ARF or RHD (or both), therefore some might have had ARF alone and no RHD? If the latter is true, the descriptor "C-RHD" (children with RHD) seems inaccurate, and the lived experience of those who were solely diagnosed with ARF is likely different from those diagnosed with RHD. I suggest further clarification, for example that all child participants had been diagnosed with RHD, if this is correct.

Thank you for this important observation. We agree that the term “ARF/RHD” in the Methods section could create ambiguity. Although the eligibility criteria permitted enrolment of children with ARF or RHD receiving secondary prophylaxis for at least six months, all participants ultimately enrolled and interviewed had a confirmed diagnosis of symptomatic RHD at the time of recruitment. No children with isolated ARF without established RHD were represented in the final study sample. This reflects the clinical context of the study sites, where children commonly present with established RHD and isolated ARF is less frequently managed within specialised services. We have revised the Results section to explicitly clarify this point and ensure that the descriptor “C-RHD” accurately reflects the study population (Page 7, Line 225-228).

8. (R2): Researcher positionality and language. Positionality is only described in terms of gender (paragraph 2, page 5), and ethnicity, nationality and language fluency are not mentioned. The paper does not clearly state the language used in the focus group discussions, or whether multiple languages were used. Transparency in this area is important to allow readers to consider possible bias, translation effects and dynamics between participants and facilitator/researchers. I suggest that the authors briefly expand on the information provided about the focus group methods, to include ethnicity of facilitators, language(s) spoken during focus groups, and whether researchers shared language(s) with participants or whether translators were used during the focus groups.

Thank you for this thoughtful comment. We agree that transparency regarding researcher positionality and language is important for interpreting potential sources of bias, translation effects, and group dynamics in qualitative research, and have incorporated this into the manuscript. Ethnicity data were not collected from participants or facilitators due to cultural sensitivity and the prevailing sociopolitical context, where explicit identification or documentation of ethnicity was considered inappropriate. All participants were Ethiopian nationals. Focus group discussions were conducted primarily in Amharic, except at Dire Dawa where discussions were conducted in Afan Oromo. Facilitators were fluent in the language used at each site and no external interpreters were involved. Audio recordings were later translated into English for analysis by a bilingual member of the research team. We also note these contextual constraints in the Discussion (Page 6, Line 187-196).

9. Results, Participants' ethnicity and language: Sociodemographic characteristics provided on children with RHD participating in the study (Results paragraph 1, page 5, and Table 1, pages 5 and 6) do not include ethnicity or primary languages spoken. It would be helpful to include these data if available to help readers' understanding of the diversity of participants.

We thank the reviewer for this important comment. Participants’ ethnicity and primary language were not collected in the sociodemographic survey. This was a deliberate decision based on cultural sensitivity and contextual considerations at the time of data collection, including heightened ethnic-based tensions and political sensitivities, such that asking participants to identify ethnicity was not considered appropriate. All child participants were Ethiopian nationals, and each study site served populations that are largely linguistically and culturally homogeneous within their respective regions. Although primary language was not systematically recorded at the individual level, routine clinical care and all study interactions (including consent and data collection) were conducted in the dominant local languages at each site (Amharic and Afan Oromo), with facilitators fluent in the language used (Page 5, Line 158-163, Page 16 Line 511-513).

10. (R2): Participant clinical characteristics: The clinical characteristics of child participants presented in Table 1, pages 5 and 6, does not provide information on whether RHD was present (see comment above referring to the use of the term "ARF/RHD" in inclusion criteria). On the assumption that all child participants did have diagnosed RHD, it would be useful to provide further subcategorization of the severity of illness, for example whether their illnesses were symptomatic or asymptomatic, if these data are available. This would help provide further context for interpreting the results, as the lived experience of RHD of those with latent disease would likely differ from those with overt illness.

Thank you for this important suggestion which we have clarified elsewhere and incorporated into the Results and Table 1 (Page 7, Line 225-236).

11. (R2): Discussion, Provider refusal to inject: In Discussion paragraph 2 (page 11) it is indicated that there was "refusal to inject in some centres". According to the results this appeared to be occurring only in one centre. I suggest that this is reworded to "...refusal to inject in one centre..."

We agree and have revised accordingly (Page 15, Line 445).

12. (R2): Supplementary 1 Text: Text Wording of questions/prompts used in focus group discussions. As described in the Methods (paragraph 1, page 5), Supplementary 1 Text provides the guide for facilitating the focus group discussions, and contains the open-ended prompts used in these discussions. The prompt provided for factors impacting prophylaxis delivery services ("What factors influence the delivery of BPG prophylaxis in healthcare settings?") uses health service wording that would not be clear to child participants with a mean age of 13.5 years. It would be helpful to provide the prompt as actually used in the focus group discussions, translated into English.

We thank the reviewer for raising this important point. We agree that the wording presented in Supplementary Text 1 reflects a structured analytic framework used by the research team, rather than the verbatim language used when engaging child participants. In the focus group discussions, facilitators did not use health-service or policy-level terminology (e.g. “structural” or “organizational” factors) with children. Instead, these domains guided the facilitators’ probing internally, while questions posed to children were phrased in simple, age-appropriate language and delivered in the local language. For example, the question listed in the supplement as “What factors influence the delivery of BPG prophylaxis in healthcare settings?” was operationalised in the discussions using child-friendly prompts such as: “Can you tell us about things that make it easy or hard for you to get your penicillin injections?”, with follow-up probes tailored to children’s responses (e.g. experiences at the clinic, interactions with nurses, pain during injections, or family support). To improve clarity and transparency, we have revised Supplementary Text 1.

All authors have approved this resubmission, and we continue to have no conflicts of interest to declare. For any correspondence regarding this manuscript, please contact me at eshetie.birru@thekids.org.au or meshetie21@gmail.com.

Sincerely,

Eshetie Melese Birru, PhD

---

## [Decision Letter · Decision Letter 1]

7 Apr 2026

PONE-D-25-52910R1Determinants of secondary prophylaxis for childhood rheumatic heart disease in Ethiopia: a qualitative study of children and caregiversPLOS One

Dear Dr. Birru,

Thank you for submitting your manuscript to PLOS ONE. After careful consideration, we feel that it has merit but does not fully meet PLOS ONE’s publication criteria as it currently stands. Therefore, we invite you to submit a revised version of the manuscript that addresses the points raised during the review process.

We look forward to receiving your revised manuscript.

Kind regards,

Dhruba Shrestha, MD

Academic Editor

PLOS One

Journal Requirements:

Reviewers' comments:

Reviewer's Responses to Questions

**Comments to the Author**

1. If the authors have adequately addressed your comments raised in a previous round of review and you feel that this manuscript is now acceptable for publication, you may indicate that here to bypass the “Comments to the Author” section, enter your conflict of interest statement in the “Confidential to Editor” section, and submit your "Accept" recommendation.

Reviewer #2: All comments have been addressed

Reviewer #3: (No Response)

2. Is the manuscript technically sound, and do the data support the conclusions?

Reviewer #2: (No Response)

Reviewer #3: No

3. Has the statistical analysis been performed appropriately and rigorously? 

Reviewer #2: (No Response)

Reviewer #3: Yes

4. Have the authors made all data underlying the findings in their manuscript fully available?

Reviewer #2: (No Response)

Reviewer #3: No

5. Is the manuscript presented in an intelligible fashion and written in standard English?

Reviewer #2: (No Response)

Reviewer #3: Yes

6. Review Comments to the Author

Reviewer #2: (No Response)

Reviewer #3: A key limitation of this paper was the presentation of the findings. There was too little description of the themes and too many quotes, leaving the reader to do all the interpretation. Achieving a balance between description (interpretation) and quotes (evidence) is crucial in qualitative research to provide both authenticity and analytical rigor. The ideal approach is to use quotes to illustrate themes, not to replace interpretation. I recommend that this section is re-written with more emphasis on the descriptions and less use and reliance on the quotes.

7. PLOS authors have the option to publish the peer review history of their article (what does this mean?). If published, this will include your full peer review and any attached files.

Reviewer #2: **Yes:** Craig Thornley

Reviewer #3: No

---

## [Author Response · Author response to Decision Letter 2]

10 Apr 2026

All comments reviewed and point by point responses attached

---

## [Decision Letter · Decision Letter 2]

5 May 2026

Determinants of secondary prophylaxis for childhood rheumatic heart disease in Ethiopia: a qualitative study of children and caregivers

PONE-D-25-52910R2

Dear Dr. Birru,

We’re pleased to inform you that your manuscript has been judged scientifically suitable for publication and will be formally accepted for publication once it meets all outstanding technical requirements.

Kind regards,

Dhruba Shrestha, MD

Academic Editor

PLOS One

Additional Editor Comments (optional):

Reviewers' comments:

Reviewer's Responses to Questions

**Comments to the Author**

1. If the authors have adequately addressed your comments raised in a previous round of review and you feel that this manuscript is now acceptable for publication, you may indicate that here to bypass the “Comments to the Author” section, enter your conflict of interest statement in the “Confidential to Editor” section, and submit your "Accept" recommendation.

Reviewer #3: All comments have been addressed

2. Is the manuscript technically sound, and do the data support the conclusions?

Reviewer #3: Yes

3. Has the statistical analysis been performed appropriately and rigorously? 

Reviewer #3: N/A

4. Have the authors made all data underlying the findings in their manuscript fully available?

Reviewer #3: No

5. Is the manuscript presented in an intelligible fashion and written in standard English?

Reviewer #3: Yes

6. Review Comments to the Author

Reviewer #3: The authors have addressed all of the prior review comments. The paper has significantly improved in the framing, presentation of findings and evidence-based discussions. It is a wonderful manuscript that makes an in-depth contribution to current literature on lived experience of ARF and RHD.

7. PLOS authors have the option to publish the peer review history of their article (what does this mean?). If published, this will include your full peer review and any attached files.

Reviewer #3: No

---

## [Editor Report · Acceptance letter]

PONE-D-25-52910R2

PLOS One

Dear Dr. Birru,

I'm pleased to inform you that your manuscript has been deemed suitable for publication in PLOS One. Congratulations! Your manuscript is now being handed over to our production team.

Kind regards,

on behalf of

Dr. Dhruba Shrestha

Academic Editor

PLOS One